# Optimizing Loss Functions Through Multivariate Taylor Polynomial Parameterization

## Abstract

Metalearning of deep neural network (DNN) architectures and hyperparameters has become an increasingly important area of research. Loss functions are a type of metaknowledge that is crucial to effective training of DNNs, however, their potential role in metalearning has not yet been fully explored. Whereas early work focused on genetic programming (GP) on tree representations, this paper proposes continuous CMA-ES optimization of multivariate Taylor polynomial parameterizations. This approach, TaylorGLO, makes it possible to represent and search useful loss functions more effectively. In MNIST, CIFAR-10, and SVHN benchmark tasks, TaylorGLO finds new loss functions that outperform the standard cross-entropy loss as well as novel loss functions previously discovered through GP, in fewer generations. These functions serve to regularize the learning task by discouraging overfitting to the labels, which is particularly useful in tasks where limited training data is available. The results thus demonstrate that loss function optimization is a productive new avenue for metalearning.

## 1 Introduction

As deep learning systems have become more complex, their architectures and hyperparameters have become increasingly difficult and time-consuming to optimize by hand. In fact, many good designs may be overlooked by humans with prior biases. Therefore, automating this process, known as metalearning, has become an essential part of the modern machine learning toolbox. Metalearning aims to solve this problem through a variety of approaches, including optimizing different aspects of the architecture from hyperparameters to topologies, and by using different methods from Bayesian optimization to evolutionary computation (Schmidhuber, 1987; Elsken et al., 2019; Miikkulainen et al., 2019; Lemke et al., 2015).

Recently, loss-function discovery and optimization has emerged as a new type of metalearning. Focusing on neural network's root training goal it aims to discover better ways to define what is being optimized. However, loss functions can be challenging to optimize because they have a discrete nested structure as well as continuous coefficients. The first system to do so, Genetic Loss Optimization (GLO; Gonzalez & Miikkulainen, 2020) tackled this problem by discovering and optimizing loss functions in two separate steps: (1) representing the structure as trees, and evolving them with Genetic Programming (GP; Banzhaf et al., 1998); and (2) optimizing the coefficients using Covariance-Matrix Adaptation Evolutionary Strategy (CMA-ES; Hansen & Ostermeier, 1996). While the approach was successful, such separate processes make it challenging to find a mutually optimal structure and coefficients. Furthermore, small changes in the tree-based search space do not always result in small changes in the phenotype, and can easily make a function invalid, making the search process ineffective.

In an ideal case, loss functions would be mapped into fixed-length vectors in a Hilbert space. This mapping should be smooth, well-behaved, well-defined, incorporate both a function's structure and coefficients, and should by its very nature exclude large classes of infeasible loss functions. This paper introduces such an approach: *Multivariate Taylor expansion-based genetic loss-function optimization* (TaylorGLO). With a novel parameterization for loss functions, the key pieces of information that affect a loss function's behavior are compactly represented in a vector. Such vectors are then optimized for a specific task using CMA-ES. Special techniques can be developed to narrow down the search space and speed up evolution.

Loss functions discovered by TaylorGLO outperform the standard cross-entropy loss (or log loss) on the MNIST, CIFAR-10, CIFAR-100, and SVHN datasets with several different network architectures. They also outperform the Baikal loss, discovered by the original GLO technique, and do it with significantly fewer function evaluations. The reason for the improved performance is that evolved functions discourage overfitting to the class labels, thereby resulting in automatic regularization. These improvements are particularly pronounced with reduced datasets where such regularization matters the most. TaylorGLO thus further establishes loss-function optimization as a promising new direction for metalearning.

## 2 RELATED WORK

Applying deep neural networks to new tasks often involves significant manual tuning of the network design. The field of metalearning has recently emerged to tackle this issue algorithmically (Schmidhuber, 1987; Lemke et al., 2015; Elsken et al., 2019; Miikkulainen et al., 2019). While much of the work has focused on hyperparameter optimization and architecture search, recently other aspects, such as activation functions and learning algorithms, have been found useful targets for optimization (Bingham et al., 2020; Real et al., 2020). Since loss functions are at the core of machine learning, it is compelling to apply metalearning to their design as well.

Deep neural networks are trained iteratively, by updating model parameters (i.e., weights and biases) using gradients propagated backward through the network (Rumelhart et al., 1985). The process starts from an error given by a loss function, which represents the primary training objective of the network. In many tasks, such as classification and language modeling, the cross-entropy loss (also known as the log loss) has been used almost exclusively. While in some approaches a regularization term (e.g. $L^2$ weight regularization; Tikhonov, 1963) is added to the the loss function definition, the core component is still the cross-entropy loss. This loss function is motivated by information theory: It aims to minimize the number of bits needed to identify a message from the true distribution, using a code from the predicted distribution.

In other types of tasks that do not fit neatly into a single-label classification framework different loss functions have been used successfully (Gonzalez et al., 2019; Gao & Grauman, 2019; Kingma & Welling, 2014; Zhou et al., 2016; Dong et al., 2017). Indeed, different functions have different properties; for instance the Huber Loss (Huber, 1964) is more resilient to outliers than other loss functions. Still, most of the time one of the standard loss functions is used without a justification; therefore, there is an opportunity to improve through metalearning.

Genetic Loss Optimization (GLO; Gonzalez & Miikkulainen, 2020) provided an initial approach into metalearning of loss functions. As described above, GLO is based on tree-based representations with coefficients. Such representations have been dominant in genetic programming because they are flexible and can be applied to a variety of function evolution domains. GLO was able to discover Baikal, a new loss function that outperformed the cross-entropy loss in image classification tasks. However, because the structure and coefficients are optimized separately in GLO, it cannot easily optimize their interactions. Many of the functions created through tree-based search are not useful because they have discontinuities, and mutations can have disproportionate effects on the functions. GLO's search is thus inefficient, requiring large populations that are evolved for many generations. Thus, GLO does not scale to the large models and datasets that are typical in modern deep learning.

The technique presented in this paper, TaylorGLO, aims to solve these problems through a novel loss function parameterization based on multivariate Taylor expansions. Furthermore, since such representations are continuous, the approach can take advantage of CMA-ES (Hansen & Ostermeier, 1996) as the search method, resulting in faster search.

## 3 LOSS FUNCTIONS AS MULTIVARIATE TAYLOR EXPANSIONS

Taylor expansions (Taylor, 1715) are a well-known function approximator that can represent differentiable functions within the neighborhood of a point using a polynomial series. Below, the common univariate Taylor expansion formulation is presented, followed by a natural extension to arbitrarily-multivariate functions.

Given a $C^{k_{\max}}$ smooth (i.e., first through $k_{\max}$ derivatives are continuous), real-valued function, $f(x) : \mathbb{R} \to \mathbb{R}$, a $k$th-order Taylor approximation at point $a \in \mathbb{R}$, $\hat{f}_k(x, a)$, where $0 \le k \le k_{\max}$, can be constructed as

$$\hat{f}_k(x, a) = \sum_{n=0}^{k} \frac{1}{n!} f^{(n)}(a)(x - a)^n. \tag{1}$$

Conventional, univariate Taylor expansions have a natural extension to arbitrarily high-dimensional inputs of $f$. Given a $C^{k_{\max}+1}$ smooth, real-valued function, $f(\mathbf{x}) : \mathbb{R}^n \to \mathbb{R}$, a $k$th-order Taylor approximation at point $\mathbf{a} \in \mathbb{R}^n$, $\hat{f}_k(\mathbf{x}, \mathbf{a})$, where $0 \le k \le k_{\max}$, can be constructed. The stricter smoothness constraint compared to the univariate case allows for the application of Schwarz's theorem on equality of mixed partials, obviating the need to take the order of partial differentiation into account.

Let us define an $n$th-degree multi-index, $\alpha = (\alpha_1, \alpha_2, \ldots, \alpha_n)$, where $\alpha_i \in \mathbb{N}_0$, $|\alpha| = \sum_{i=1}^{n} \alpha_i$, $\alpha! = \prod_{i=1}^{n} \alpha_i!$. $\mathbf{x}^\alpha = \prod_{i=1}^{n} x_i^{\alpha_i}$, and $\mathbf{x} \in \mathbb{R}^n$. Multivariate partial derivatives can be concisely written using a multi-index

$$\partial^\alpha f = \partial_1^{\alpha_1} \partial_2^{\alpha_2} \cdots \partial_n^{\alpha_n} f = \frac{\partial^{|\alpha|}}{\partial x_1^{\alpha_1} \partial x_2^{\alpha_2} \cdots \partial x_n^{\alpha_n}}. \tag{2}$$

Thus, discounting the remainder term, the multivariate Taylor expansion for $f(\mathbf{x})$ at $\mathbf{a}$ is

$$\hat{f}_k(\mathbf{x}, \mathbf{a}) = \sum_{\forall \alpha, |\alpha| \le k} \frac{1}{\alpha!} \partial^\alpha f(\mathbf{a})(\mathbf{x} - \mathbf{a})^\alpha. \tag{3}$$

The unique partial derivatives in $\hat{f}_k$ and $\mathbf{a}$ are parameters for a $k$th order Taylor expansion. Thus, a $k$th order Taylor expansion of a function in $n$ variables requires $n$ parameters to define the center, $\mathbf{a}$, and one parameter for each unique multi-index $\alpha$, where $|\alpha| \le k$. That is: $\#_{\text{parameters}}(n, k) = n + \binom{n+k}{k} = n + \frac{(n+k)!}{n! \, k!}$.

The multivariate Taylor expansion can be leveraged for a novel loss-function parameterization. Let an $n$-class classification loss function be defined as $\mathcal{L}_{\text{Log}} = -\frac{1}{n} \sum_{i=1}^{n} f(x_i, y_i)$. The function $f(x_i, y_i)$ can be replaced by its $k$th-order, bivariate Taylor expansion, $\hat{f}_k(x, y, a_x, a_y)$. More sophisticated loss functions can be supported by having more input variables beyond $x_i$ and $y_i$, such as a time variable or unscaled logits. This approach can be useful, for example, to evolve loss functions that change as training progresses.

For example, a loss function in $\mathbf{x}$ and $\mathbf{y}$ has the following third-order parameterization with parameters $\theta$ (where $\mathbf{a} = \langle \theta_0, \theta_1 \rangle$):

$$\begin{aligned}
\mathcal{L}(\mathbf{x}, \mathbf{y}) = -\frac{1}{n} \sum_{i=1}^{n} \Big[ & \theta_2 + \theta_3(y_i - \theta_1) + \tfrac{1}{2}\theta_4(y_i - \theta_1)^2 + \tfrac{1}{6}\theta_5(y_i - \theta_1)^3 + \theta_6(x_i - \theta_0) \\
& + \theta_7(x_i - \theta_0)(y_i - \theta_1) + \tfrac{1}{2}\theta_8(x_i - \theta_0)(y_i - \theta_1)^2 + \tfrac{1}{2}\theta_9(x_i - \theta_0)^2 \\
& + \tfrac{1}{2}\theta_{10}(x_i - \theta_0)^2(y_i - \theta_1) + \tfrac{1}{6}\theta_{11}(x_i - \theta_0)^3 \Big]
\end{aligned} \tag{4}$$

Notably, the reciprocal-factorial coefficients can be integrated to be a part of the parameter set by direct multiplication if desired.

As will be shown in this paper, the technique makes it possible to train neural networks that are more accurate and learn faster than those with tree-based loss function representations. Representing loss functions in this manner confers several useful properties:

- It guarantees smooth functions;
- Functions do not have poles (i.e., discontinuities going to infinity or negative infinity) within their relevant domain;
- They can be implemented purely as compositions of addition and multiplication operations;
- They can be trivially differentiated;
- Nearby points in the search space yield similar results (i.e., the search space is locally smooth), making the fitness landscape easier to search;
- Valid loss functions can be found in fewer generations and with higher frequency;
- Loss function discovery is consistent and not dependent on a specific initial population; and
- The search space has a tunable complexity parameter (i.e., the order of the expansion).

These properties are not necessarily held by alternative function approximators. For instance:

**Fourier series** are well suited for approximating periodic functions (Fourier, 1829). Consequently, they are not as well suited for loss functions, whose local behavior within a narrow domain is important. Being a composition of waves, Fourier series tend to have many critical points within the domain of interest. Gradients fluctuate around such points, making gradient descent infeasible. Additionally, close approximations require a large number of terms, which in itself can be injurious, causing large, high-frequency fluctuations known as "ringing", due to Gibb's phenomenon (Wilbraham, 1848).

**Padé approximants** can be more accurate approximations than Taylor expansions; indeed, Taylor expansions are a special case of Padé approximants where $M = 0$ (Graves-Morris, 1979). However, unfortunately Padé approximants can model functions with one or more poles, which valid loss functions typically should not have. These problems still exist, and are exacerbated, for Chisholm approximants (a bivariate extension; Chisholm, 1973) and Canterbury approximants (a multivariate generalization; Graves-Morris & Roberts, 1975).

**Laurent polynomials** can represent functions with discontinuities, the simplest being $x^{-1}$. While Laurent polynomials provide a generalization of Taylor expansions into negative exponents, the extension is not useful because it results in the same issues as Padé approximants.

**Polyharmonic splines** can represent continuous functions within a finite domain, however, the number of parameters is prohibitive in multivariate cases.

The multivariate Taylor expansion is therefore a better choice than the alternatives. It makes it possible to optimize loss functions efficiently in TaylorGLO, as will be described next.

## 4 THE TAYLORGLO METHOD

TaylorGLO (Figure 1) aims to find the optimal parameters for a loss function represented as a multivariate Taylor expansion. The parameters for a Taylor approximation (i.e., the center point and partial derivatives) are referred to as $\theta_{\hat{f}}$: $\theta_{\hat{f}} \in \Theta$, $\Theta = \mathbb{R}^{\#\text{parameters}}$. TaylorGLO strives to find the vector $\theta_{\hat{f}}^*$ that parameterizes the optimal loss function for a task. Because the values are continuous, as opposed to discrete graphs of the original GLO, it is possible to use continuous optimization methods.

In particular, Covariance Matrix Adaptation Evolutionary Strategy (CMA-ES Hansen & Ostermeier, 1996) is a popular population-based, black-box optimization technique for rugged, continuous spaces. CMA-ES functions by maintaining a covariance matrix around a mean point that represents a distribution of solutions. At each generation, CMA-ES adapts the distribution to better fit evaluated objective values from sampled

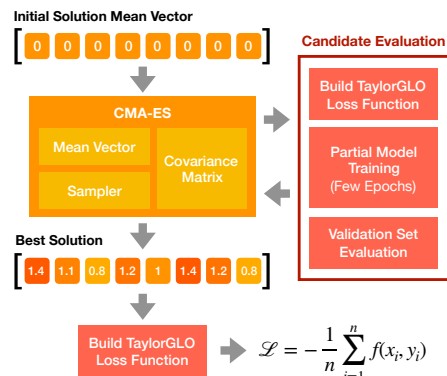

Figure 1: The TaylorGLO method. Starting with a population of initially unbiased loss functions, CMA-ES optimizes their Taylor expansion parameters in order to maximize validation accuracy after partial training. The candidate with the highest accuracy is chosen as the final, best solution.

individuals. In this manner, the area in the search space that is being sampled at each step grows, shrinks, and moves dynamically as needed to maximize sampled candidates' fitnesses. TaylorGLO uses the $(\mu/\mu, \lambda)$ variant of CMA-ES (Hansen & Ostermeier, 2001), which incorporates weighted rank-$\mu$ updates (Hansen & Kern, 2004) to reduce the number of objective function evaluations needed.

In order to find $\theta_{\hat{f}}^*$, at each generation CMA-ES samples points in $\Theta$. Their fitness is determined by training a model with the corresponding loss function and evaluating the model on a validation dataset. Fitness evaluations may be distributed across multiple machines in parallel and retried a limited number of times upon failure. An initial vector of $\theta_{\hat{f}} = \mathbf{0}$ is chosen as a starting point in the search space to avoid bias.

Fully training a model can be prohibitively expensive in many problems. However, performance near the beginning of training is usually correlated with performance at the end of training, and therefore it is enough to train the models only partially to identify the most promising candidates. This type of

approximate evaluation is common in metalearning (Grefenstette & Fitzpatrick, 1985; Jin, 2011). An additional positive effect is that evaluation then favors loss functions that learn more quickly.

For a loss function to be useful, it must have a derivative that depends on the prediction. Therefore, internal terms that do not contribute to $\frac{\partial}{\partial \mathbf{y}} \mathcal{L}_f(\mathbf{x}, \mathbf{y})$ can be trimmed away. This step implies that any term $t$ within $f(x_i, y_i)$ with $\frac{\partial}{\partial y_i} t = 0$ can be replaced with 0. For example, this refinement simplifies Equation 4, providing a reduction in the number of parameters from twelve to eight:

$$\mathcal{L}(\mathbf{x}, \mathbf{y}) = -\frac{1}{n} \sum_{i=1}^{n} \left[ \theta_2 (y_i - \theta_1) + \frac{1}{2}\theta_3 (y_i - \theta_1)^2 + \frac{1}{6}\theta_4 (y_i - \theta_1)^3 + \theta_5 (x_i - \theta_0)(y_i - \theta_1) \right.$$
$$\left. + \frac{1}{2}\theta_6 (x_i - \theta_0)(y_i - \theta_1)^2 + \frac{1}{2}\theta_7 (x_i - \theta_0)^2 (y_i - \theta_1) \right]. \tag{5}$$

## 5 EXPERIMENTAL SETUP

This section presents the experimental setup that was used to evaluate the TaylorGLO technique.

**Domains:** MNIST (LeCun et al., 1998) was included as simple domain to illustrate the method and to provide a backward comparison with GLO; CIFAR-10 (Krizhevsky & Hinton, 2009), CIFAR-100 (Krizhevsky & Hinton, 2009), and SVHN (Netzer et al., 2011) were included as more modern benchmarks. Improvements were measured in comparison to the standard cross-entropy loss function $\mathcal{L}_{\text{Log}} = -\frac{1}{n} \sum_{i=1}^{n} x_i \log(y_i)$, where $x$ is sampled from the true distribution, $y$ is from the predicted distribution, and $n$ is the number of classes.

**Evaluated architectures:** A variety of architectures were used to evaluate TaylorGLO: the basic CNN architecture evaluated in the GLO study (Gonzalez & Miikkulainen, 2020), AlexNet (Krizhevsky et al., 2012), AllCNN-C (Springenberg et al., 2015), Preactivation ResNet-20 (He et al., 2016a), which is an improved variant of the ubiquitous ResNet architecture (He et al., 2016b), and Wide ResNets of different morphologies (Zagoruyko & Komodakis, 2016). Networks with Cutout (DeVries & Taylor, 2017) and CutMix (Yun et al., 2019) were also evaluated, to show that TaylorGLO provides a different, complementary approach to regularization.

**TaylorGLO setup:** CMA-ES was instantiated with population size $\lambda = 28$ on MNIST and $\lambda = 20$ on all other datasets, and an initial step size $\sigma = 1.2$. These values were found to work well in preliminary experiments. The candidates were third-order (i.e., $k = 3$) TaylorGLO loss functions (Equation 5). Such functions were found experimentally to have a better trade-off between evolution time and performance compared to second- and fourth-order TaylorGLO loss functions, although the differences were relatively small.

Further experimental setup and implementation details are provided in Appendix A.

## 6 RESULTS

This section illustrates the TaylorGLO process and demonstrates how the evolved loss functions can improve performance over the standard cross-entropy loss function, especially on reduced datasets. A summary of results on three datasets across a variety of models are shown in Table 1.

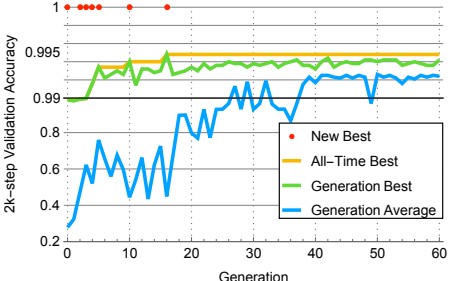

Figure 2: The process of discovering loss functions in MNIST. Red dots mark generations where new improved loss functions were found. TaylorGLO discovers good functions in very few generations. The best had a 2000-step validation accuracy of 0.9948, compared to 0.9903 with the cross-entropy loss, averaged over ten runs. This difference translates to a similar improvement on the test set, as shown in Table 1.

### 6.1 THE TAYLORGLO DISCOVERY PROCESS

Figure 2 illustrates the evolution process over 60 generations, which is sufficient to reach convergence on the MNIST dataset. TaylorGLO is able to discover highly-performing loss functions quickly, i.e. within 20 generations. Generations' average validation accuracy approaches generations' best accuracy as evolution progresses, indicating that population as a whole is improving. Whereas

Table 1: Test-set accuracy of loss functions discovered by TaylorGLO compared with that of the cross-entropy loss. The TaylorGLO results are based on the loss function with the highest validation accuracy during evolution. All averages are from ten separately trained models and $p$-values are from one-tailed Welch's $t$-Tests. Standard deviations are shown in parentheses. TaylorGLO discovers loss functions that perform significantly better than the cross-entropy loss in almost all cases, including those that include Cutout, suggesting that it provides a different form of regularization.

| Task and Model | Avg. TaylorGLO Acc. | Avg. Baseline Acc. | $p$-value |
|---|---|---|---|
| **MNIST** | | | |
| Basic CNN [1] | **0.9951 (0.0005)** | 0.9899 (0.0003) | $2.95 \times 10^{-15}$ |
| **CIFAR-10** | | | |
| AlexNet [2] | **0.7901 (0.0026)** | 0.7638 (0.0046) | $1.76 \times 10^{-10}$ |
| AlexNet + Cutout [6] | **0.7786 (0.0022)** | 0.7741 (0.0040) | 0.0049 |
| AlexNet + CutMix [7] | **0.7928 (0.0027)** | 0.7856 (0.0026) | $8.13 \times 10^{-6}$ |
| PreResNet-20 [4] | **0.9169 (0.0014)** | 0.9153 (0.0021) | 0.0400 |
| AllCNN-C [3] | **0.9271 (0.0013)** | 0.8965 (0.0021) | $0.42 \times 10^{-17}$ |
| AllCNN-C [3] + Cutout [6] | **0.9329 (0.0022)** | 0.8911 (0.0037) | $1.60 \times 10^{-14}$ |
| AllCNN-C [3] + CutMix [7] | **0.9327 (0.0014)** | 0.8749 (0.0042) | $1.89 \times 10^{-13}$ |
| Wide ResNet 16-8 [5] | **0.9558 (0.0011)** | 0.9528 (0.0012) | $1.77 \times 10^{-5}$ |
| Wide ResNet 16-8 [5] + Cutout [6] | **0.9618 (0.0010)** | 0.9582 (0.0011) | $2.55 \times 10^{-7}$ |
| Wide ResNet 28-5 [5] | 0.9548 (0.0015) | 0.9556 (0.0011) | 0.0984 |
| Wide ResNet 28-5 [5] + Cutout [6] | 0.9621 (0.0013) | 0.9616 (0.0011) | 0.1882 |
| **CIFAR-100** | | | |
| PyramidNet 110a48 [8] | 0.7409 (0.0040) | **0.7523 (0.0037)** | $3.87 \times 10^{-6}$ |
| PyramidNet 110a48 [8] + Cutout [6] | **0.7708 (0.0029)** | 0.7674 (0.0036) | 0.0189 |
| **SVHN** | | | |
| Wide ResNet 16-8 [5] | **0.9658 (0.0007)** | 0.9597 (0.0006) | $1.94 \times 10^{-13}$ |
| Wide ResNet 16-8 [5] + Cutout [6] | **0.9714 (0.0010)** | 0.9673 (0.0008) | $9.10 \times 10^{-9}$ |
| Wide ResNet 28-5 [5] | **0.9657 (0.0009)** | 0.9634 (0.0006) | $6.62 \times 10^{-6}$ |
| Wide ResNet 28-5 [5] + Cutout [6] | **0.9727 (0.0006)** | 0.9709 (0.0006) | $2.96 \times 10^{-6}$ |

Network architecture references: [1] Gonzalez & Miikkulainen (2020)  [2] Krizhevsky et al. (2012)  [3] Springenberg et al. (2015)  [4] He et al. (2016a)  [5] Zagoruyko & Komodakis (2016)  [6] DeVries & Taylor (2017)  [7] Yun et al. (2019)  [8] Han et al. (2017)

GLO's unbounded search space often results in pathological functions, every TaylorGLO training session completed successfully without any instabilities.

Figure 3 shows the shapes and parameters of each generation's highest-scoring loss function. In Figure 3$a$ the functions are plotted as if they were being used for binary classification, i.e. the loss for an incorrect label on the left and for a correct one on the right (Gonzalez & Miikkulainen, 2020). The functions have a distinct pattern through the evolution process. Early generations include a wider variety of shapes, but they later converge towards curves with a shallow minimum around $y_0 = 0.8$. In other words, the loss increases near the correct output—which is counterintuitive. This shape is also strikingly different from the cross-entropy loss, which decreases monotonically from left to right, as one might expect all loss functions to do. The evolved shape is effective most likely because can provide an implicit regularization effect: it discourages the model from outputting unnecessarily extreme values for the correct class, and therefore makes overfitting less likely (Gonzalez & Miikkulainen, 2020). This is a surprising finding, and demonstrates the power of machine learning to create innovations beyond human design.

## 6.2 PERFORMANCE COMPARISONS

Over 10 fully-trained models, the best TaylorGLO loss function achieved a mean testing accuracy of **0.9951** (stddev 0.0005) in MNIST. In comparison, the cross-entropy loss only reached 0.9899 (stddev 0.0003), and the "BaikalCMA" loss function discovered by GLO, 0.9947 (stddev 0.0003) (Gonzalez & Miikkulainen, 2020); both differences are statistically significant (Figure 5). Notably, TaylorGLO achieved this result with significantly fewer generations. GLO required 11,120 partial evaluations (i.e., 100 individuals over 100 GP generations plus 32 individuals over 35 CMA-ES generations),

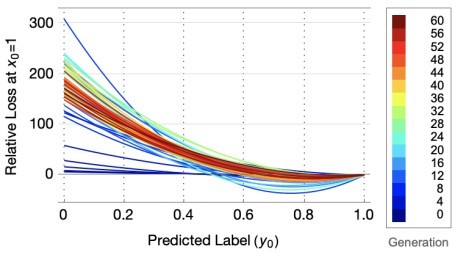

(a) Best discovered functions over time

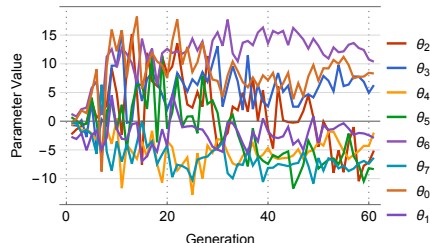

(a) Best function parameters over time

Figure 3: The best loss functions (a) and their respective parameters (b) from each generation of TaylorGLO on MNIST. The functions are plotted in a binary classification modality, showing loss for different values of the network output ($y_0$ in the horizontal axis) when the correct label is 1.0. The functions are colored according to their generation from blue to red, and vertically shifted such that their loss at $y_0 = 1$ is zero (the raw value of a loss function is not relevant; the derivative, however, is). TaylorGLO explores varying shapes of solutions before narrowing down on functions in the red band; this process can also be seen in (b), where parameters become more consistent over time, and in the population plot of Appendix B. The final functions decrease from left to right, but have a significant increase in the end. This shape is likely to prevent overfitting during learning, which leads to the observed improved accuracy.

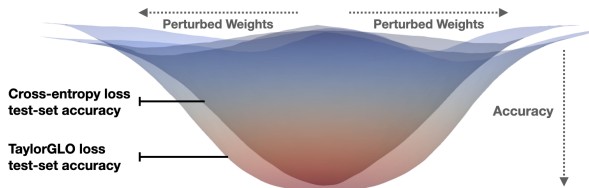

Figure 4: Accuracy basins for AllCNN-C models trained with both cross-entropy and TaylorGLO loss functions. The TaylorGLO basins are both flatter and lower, indicating that they are more robust and generalize better (Keskar et al., 2017), which results in higher accuracy.

while the top TaylorGLO loss function only required **448** partial evaluations, i.e. 4.03% as many. Thus, TaylorGLO achieves improved results with significantly fewer evaluations than GLO.

Due to the very large number evaluations required by GLO, TaylorGLO is only compared to GLO on MNIST. GLO is not practically applicable to deeper models with longer training times. For example, even a relatively small deep network, PreResNet-20 He et al. (2016a), would require over 171 GPU days of computation, assuming the same number of evaluations as above on MNIST.

The large reduction in evaluations during evolution compared to GLO allows TaylorGLO to tackle harder problems, including models that have millions of parameters. On CIFAR-10, CIFAR-100, and SVHN, TaylorGLO was able to outperform cross-entropy baselines consistently on a variety models, as shown in Table 1. These increases in accuracy are greater than what is possible through implicit learning rate adjustment alone (detailed in Appendix E). TaylorGLO also provides further improvement on architectures that use Cutout (DeVries & Taylor, 2017), suggesting that its mechanism of avoiding overfitting is different from other regularization techniques.

In addition, TaylorGLO loss functions result in more robust trained models. In Figure 4, accuracy basins for two AllCNN-C models, one trained with the TaylorGLO loss function and another with the cross-entropy loss, are plotted along a two-dimensional slice $[-1, 1]$ of the weight space (a technique due to Li et al., 2018). The TaylorGLO loss function results in a flatter, lower basin. This result suggests that the model is more robust, i.e. its performance is less sensitive to small perturbations in the weight space, and it also generalizes better (Keskar et al., 2017).

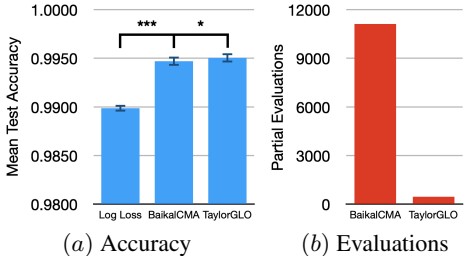
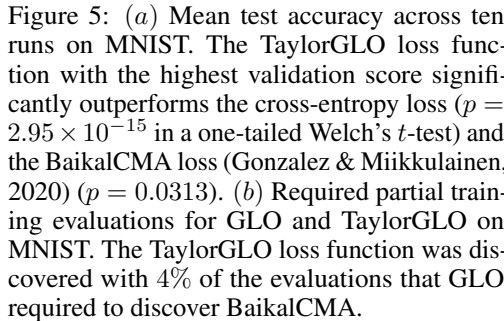

(a) Accuracy      (b) Evaluations

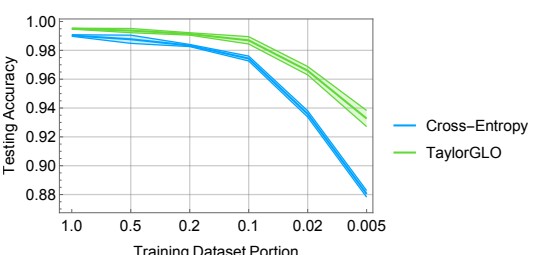

Figure 5: (a) Mean test accuracy across ten runs on MNIST. The TaylorGLO loss function with the highest validation score significantly outperforms the cross-entropy loss ($p = 2.95 \times 10^{-15}$ in a one-tailed Welch's $t$-test) and the BaikalCMA loss (Gonzalez & Miikkulainen, 2020) ($p = 0.0313$). (b) Required partial training evaluations for GLO and TaylorGLO on MNIST. The TaylorGLO loss function was discovered with $4\%$ of the evaluations that GLO required to discover BaikalCMA.

Figure 6: Accuracy with reduced portions of the MNIST dataset. Progressively smaller portions of the dataset were used to train the models (averaging over ten runs). The TaylorGLO loss function provides significantly better performance than the cross-entropy loss on all training dataset sizes, and particularly on the smaller datasets. Thus, its ability to discourage overfitting is particularly useful in applications where only limited data is available.

## 6.3 PERFORMANCE ON REDUCED DATASETS

The performance improvements that TaylorGLO provides are especially pronounced with reduced datasets. For example, Figure 6 compares accuracies of models trained for 20,000 steps on different portions of the MNIST dataset (similar results were obtained with other datasets and architectures). Overall, TaylorGLO significantly outperforms the cross-entropy loss. When evolving a TaylorGLO loss function and training against $10\%$ of the training dataset, with 225 epoch evaluations, TaylorGLO reached an average accuracy across ten models of **0.7595** (stddev 0.0062). In contrast, only four out of ten cross-entropy loss models trained successfully, with those reaching a lower average accuracy of 0.6521. Thus, customized loss functions can be especially useful in applications where only limited data is available to train the models, presumably because they are less likely to overfit to the small number of examples.

## 7 DISCUSSION AND FUTURE WORK

TaylorGLO was applied to the benchmark tasks using various standard architectures with standard hyperparameters. These setups have been heavily engineered and manually tuned by the research community, yet TaylorGLO was able to improve them. Interestingly, the improvements were more substantial with wide architectures and smaller with narrow and deep architectures such as the Preactivation ResNet. While it may be possible to further improve upon this result, it is also possible that loss function optimization is more effective with architectures where the gradient information travels through fewer connections, or is otherwise better preserved throughout the network. An important direction of future work is therefore to evolve both loss functions and architectures together, taking advantage of possible synergies between them.

As illustrated in Figure 3a, the most significant effect of evolved loss functions is to discourage extreme output values, thereby avoiding overfitting. It is interesting that this mechanism is apparently different from other regularization techniques such as dropout (as shown by Gonzalez & Miikkulainen, 2020) and data augmentation with Cutout (as seen in Table 1). Dropout and Cutout improve performance over the baseline, and loss function optimization improves it further. This result suggests that regularization is a multifaceted process, and further work is necessary to understand how to best take advantage of it.

Another important direction is to incorporate state information into TaylorGLO loss functions, such as the percentage of training steps completed. TaylorGLO may then find loss functions that are best suited for different points in training, where, for example, different kinds of regularization work best (Golatkar et al., 2019). Unintuitive changes to the training process, such as cycling learning rates

(Smith, 2017), have been found to improve performance; evolution could be used to find other such opportunities automatically. Batch statistics could help evolve loss functions that are more well-tuned to each batch; intermediate network activations could expose information that may help tune the function for deeper networks like ResNet. Deeper information about the characteristics of a model's weights and gradients, such as that from spectral decomposition of the Hessian matrix (Sagun et al., 2017), could assist the evolution of loss functions that adapt to the current fitness landscape. The technique could also be adapted to models with auxiliary classifiers (Szegedy et al., 2015) as a means to touch deeper parts of the network.

## 8 CONCLUSION

This paper proposes TaylorGLO as a promising new technique for loss-function metalearning. TaylorGLO leverages a novel parameterization for loss functions, allowing the use of continuous optimization rather than genetic programming for the search, thus making it more efficient and more reliable. TaylorGLO loss functions serve to regularize the learning task, outperforming the standard cross-entropy loss significantly on MNIST, CIFAR-10, CIFAR-100, and SVHN benchmark tasks with a variety of network architectures. They also outperform previously loss functions discovered in prior work, while requiring many fewer candidates to be evaluated during search. Thus, TaylorGLO results in higher testing accuracies, better data utilization, and more robust models, and is a promising new avenue for metalearning.

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

## A    EXPERIMENTAL SETUP

The following subsections cover specific experimental setup details. The three evaluated datasets are detailed in how they were used, along with implementation details.

### A.1    MNIST

The first domain was MNIST Handwritten Digits, a widely used dataset where the goal is to classify $28 \times 28$ pixel images as one of ten digits. MNIST has 55,000 training samples, 5,000 validation samples, and 10,000 testing samples. The dataset is well understood and relatively quick to train, and forms a good foundation for understanding how TaylorGLO evolves loss functions.

The basic CNN architecture evaluated in the GLO study (Gonzalez & Miikkulainen, 2020) can also be used to provide a direct point of comparison with prior work on MNIST. Importantly, this architecture includes a dropout layer (Hinton et al., 2012) for explicit regularization. As in GLO, training is based on stochastic gradient descent (SGD) with a batch size of 100, a learning rate of 0.01, and, unless otherwise specified, occurred over 20,000 steps.

### A.2    CIFAR-10 AND CIFAR-100

To validate TaylorGLO in a more challenging context, the CIFAR-10 (Krizhevsky & Hinton, 2009) dataset was used. It consists of small $32 \times 32$ pixel color photographs of objects in ten classes. CIFAR-10 traditionally consists of 50,000 training samples, and 10,000 testing samples; however 5,000 samples from the training dataset were used for validation of candidates, resulting in 45,000 training samples.

Models were trained with their respective hyperparameters from the literature. Inputs were normalized by subtracting their mean pixel value and dividing by their pixel standard deviation. Standard data augmentation techniques consisting of random horizontal flips and croppings with two pixel padding were applied during training.

CIFAR-100 is a similar, though significantly more challenging, dataset where a different set of 60,000 images is divided into 100 classes, instead of 10. The same splits for training, validation, and testing were used for CIFAR-100 as for CIFAR-10, and evaluate TaylorGLO further.

### A.3    SVHN

The Street View House Numbers (SVHN; Netzer et al., 2011) dataset is another image classification domain that was used to evaluate TaylorGLO, consisting of $32 \times 32$ pixel images of numerical digits from Google Street View. SVHN consists of 73,257 training samples, 26,032 testing samples, and 531,131 supplementary, easier training samples. To reduce computation costs, supplementary examples were not used during training; this fact explains why presented baselines may be lower than other SVHN baselines in the literature. Since a validation set is not in the standard splits, 26,032 samples from the training dataset were used for validation of candidates, resulting in 47,225 training samples.

As with CIFAR-10, models were trained with their respective hyperparameters from the literature and with the same data augmentation pipeline.

### A.4    CANDIDATE EVALUATION DETAILS

During candidate evaluation, models were trained for 10% of a full training run on MNIST, equal to 2,000 steps (i.e., four epochs). An in-depth analysis on the technique's sensitivity to training steps during candidate evaluation is provided in Appendix D—overall, the technique is robust even with few training steps. However, on more complex models with abrupt learning rate decay schedules, greater numbers of steps provide better fitness estimates.

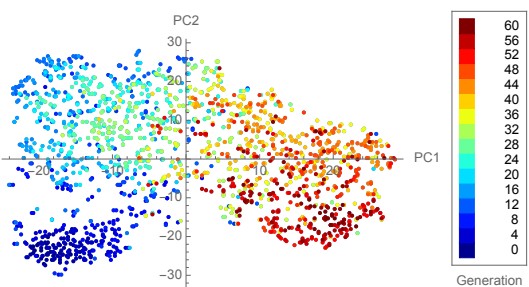

Figure 7: A visualization of all TaylorGLO loss function candidates using t-SNE (Maaten & Hinton, 2008) on MNIST. Colors map to each candidate's generation. Loss function populations show an evolutionary path and focus over time towards functions that perform well, consistent with the convergence and settling in Figure 3.

A.5 STATISTICAL TESTING

Statistical significance tests define a null hypothesis and reject it if a $p$-value is below a predefined significance level, typically 0.05. A $p$-value is the probability of obtaining extreme results at the same level or greater than the results observed given that the null hypothesis is true.

When comparing results in this paper, a one-tailed null hypothesis is typically used:

$$H_0 : \neg\left(\mu_1 < \mu_2\right), \tag{6}$$

where $\mu_1$ and $\mu_2$ are mean values from two separate sets of training sessions. The rejection of this null hypothesis implies that $\mu_2$ is statistically significantly larger than $\mu_1$. Thus, the change between the two sets of training sessions is robust to training stochasticity, such as from varying weight initializations.

Throughout this paper, Welch's $t$-Test Welch (1947) is used to determine statistical significance when comparing sets of results which may not have equal variances. It is also a better fit than Student's $t$-Test due to its higher robustness and statistical power Ruxton (2006).

A.6 IMPLEMENTATION DETAILS

Due to the number of partial training sessions that are needed to evaluate TaylorGLO loss function candidates, training was distributed across the network to a cluster, composed of dedicated machines with NVIDIA GeForce GTX 1080Ti GPUs. Training itself was implemented with TensorFlow (Abadi et al., 2016) in Python. The primary components of TaylorGLO (i.e., the genetic algorithm and CMA-ES) were implemented in the Swift programming language which allows for easy parallelization. These components run centrally on one machine and asynchronously dispatch work to the cluster.

Training for each candidate was aborted and retried up to two additional times if validation accuracy was below 0.15 at the tenth epoch. This method helped reduce computation costs.

B ILLUSTRATING THE EVOLUTIONARY PROCESS

The TaylorGLO search process can be illustrated with t-SNE dimensionality reduction (Maaten & Hinton, 2008) on *every* candidate loss function within a run (Figure 7). The initial points (i.e. loss functions) are initially widespread on the left side, but quickly migrate and spread to the right as CMA-ES explores the parameter space, and eventually concentrate in a smaller region of dark red points. This pattern is consistent with the convergence and settling in Figure 3.

C TOP MNIST LOSS FUNCTION

The best loss function obtained from running TaylorGLO on MNIST was found in generation 74. This function, with parameters $\theta = \langle 11.9039, -4.0240, 6.9796, 8.5834, -1.6677, 11.6064, 12.6684, -3.4674\rangle$ (rounded to four decimal-places), achieved a 2k-step validation accuracy of 0.9950 on its

single evaluation, higher than 0.9903 for the cross entropy loss. This loss function was a modest improvement over the previous best loss function from generation 16, which had a validation accuracy of 0.9958.

## D   MNIST EVALUATION LENGTH SENSITIVITY

**200-step**   TaylorGLO is surprisingly resilient when evaluations during evolution are shortened to 200 steps (i.e., 0.4 epochs) of training. With so little training, returned accuracies are noisy and dependent on each individual network's particular random initialization. On a 60-generation run with 200-step evaluations, the best evolved loss function had a mean testing accuracy of 0.9946 across ten samples, with a standard deviation of 0.0016. While slightly lower, and significantly more variable, than the accuracy for the best loss function that was found on the main 2,000-step run, the accuracy is still significantly higher than that of the cross-entropy baseline, with a $p$-value of $6.3 \times 10^{-6}$. This loss function was discovered in generation 31, requiring 1,388.8 2,000-step-equivalent partial evaluations. That is, evolution with 200-step partial evaluations is over three-times less sample efficient than evolution with 2,000-step partial evaluations.

**20,000-step**   On the other extreme, where evaluations consist of the same number of steps as a full training session, one would expect better loss functions to be discovered, and more reliably, because the fitness estimates are less noisy. Surprisingly, that is not the case: The best loss function had a mean testing accuracy of 0.9945 across ten samples, with a standard deviation of 0.0015. While also slightly lower, and also significantly more variable, than the accuracy for the best loss function that was found on the main 2,000-step run, the accuracy is significantly higher than the cross-entropy baseline, with a $p$-value of $5.1 \times 10^{-6}$. This loss function was discovered in generation 45, requiring 12,600 2,000-step-equivalent partial evaluations. That is, evolution with 20,000-step full evaluations is over 28-times less sample efficient than evolution with 2,000-step partial evaluations.

These results thus suggest that there is an optimal way to evaluate candidates during evolution, resulting in lower computational cost and better loss functions. Notably, the best evolved loss functions from all three runs (i.e., 200-, 2,000-, and 20,000-step) have similar shapes, reinforcing the idea that partial-evaluations can provide useful performance estimates.

## E   LEARNING RATE SENSITIVITY

Loss functions can embody different learning rates implicitly. This section shows that TaylorGLO loss functions' benefits come from more than just metalearning such learning rates. Increases in performance that result from altering the base learning rate with cross-entropy loss are significantly smaller than those that TaylorGLO provides.

More specifically, Figure 8 quantifies the effect of varying learning rates on the final testing accuracy of AllCNN-C models trained on CIFAR-10. AllCNN-C was chosen for this analysis since it exhibits the largest variations in performance, making this effect more clear. While learning rates larger than 0.01 (the standard learning rate for AllCNN-C) reach slightly higher accuracies, this effect comes at the cost of less stable training. The majority of models trained with these higher learning rates failed to train. Thus, the standard choice of learning rate for AllCNN-C is appropriate for the cross-entropy loss, and TaylorGLO loss functions are able to improve upon it.

## F   TAYLOR APPROXIMATIONS OF THE CROSS-ENTROPY LOSS

While TaylorGLO's performance originates primarily from discovering better loss functions, it is informative to analyze what role the accuracy of the Taylor approximation plays in it. One way to characterize this effect is to analyze the performance of various Taylor approximations of the cross-entropy loss.

Table 2 provides results from such a study. Bivariate approximations to the cross-entropy loss, centered at $\mathbf{a} = \langle 0.5, 0.5 \rangle$, with different orders $k$ were used to train AllCNN-C models on CIFAR-10. Third-order approximations and above are trainable. Approximations' performance is within a few

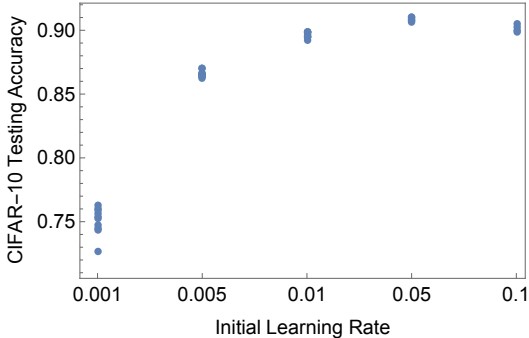

Figure 8: Effect of varying learning rates in AllCNN-C when trained with the cross-entropy loss on CIFAR-10. For each learning rate, ten models were trained, with up to ten retries if training failed. The majority of training attempts failed for learning rates larger than 0.01. The 0.01 learning rate used in the experiments in this paper results in best stable performance. Overall, the small performance differences that can result from adjusting the learning rate, regardless of stability, are much smaller than those that result from training with TaylorGLO. Thus, TaylorGLO provides a mechanism for improvement beyond implicit adjustments of the learning rate.

Table 2: Performance of Taylor approximations of the cross-entropy loss function on AllCNN-C with CIFAR-10. Approximations of different orders, with $\mathbf{a} = \langle 0.5, 0.5 \rangle$, are presented. Presented accuracies are the mean from ten runs. The baseline is the standard cross-entropy loss. Higher-order approximations are better, suggesting a potential (although computationally expensive) opportunity for improvement in the future.

| Loss Function | Mean Accuracy (stddev) |
|---|---|
| $k = 2$ | 0.1034 (0.0101) |
| $k = 3$ | 0.8451 (0.0043) |
| $k = 4$ | 0.8592 (0.0032) |
| $k = 5$ | 0.8649 (0.0042) |
| Cross-Entropy | **0.8965 (0.0021)** |

percentage points of the cross-entropy loss, with higher-order approximations yielding progressively better accuracies, as expected.

The results thus show that third-order TaylorGLO loss functions cannot represent the cross-entropy baseline loss accurately. One possibility for improving TaylorGLO is thus to utilize higher order approximations. However, it is remarkable that TaylorGLO can still find loss functions that outperform the cross-entropy loss. Also, the increase in the number of parameters—and the corresponding increase in computational requirements—may in practice outweigh the benefits from a finer-grained representation. This effect was seen in preliminary experiments, and the third-order approximations (used in this paper) deemed to strike a good balance.

## G  TAYLORGLO EXPERIMENT DURATIONS AND ENVIRONMENTAL IMPACT

The infrastructure that ran the experiments in this paper is located in California, which is estimated to have had an estimated carbon dioxide equivalent total output emission rate of 226.21 kgCO$_2$eq/kWh in 2018 (epa, 2020). This quantity can be used to calculate the climate impact of compute-intensive experiments.

Table 3 provides estimates of durations and total emissions for various TaylorGLO experiments. Emissions were calculated using the Machine Learning Impact calculator (Lacoste et al., 2019), assuming that no candidates failed evaluation (which would result in slightly lower estimates). Presented values can thus be thought of as being an upper bound.

Table 3: Estimated TaylorGLO experiment durations and total emissions. The estimates assume populations of 20 concurrent candidates and 50 generation runs. Emission values are upper bounds reported in equivalent kilograms of carbon dioxide, thus accounting for other gases of interest. Overall, experiments are short enough that they can each be run over a few days.

| TaylorGLO Experiment | Duration (hours) | Total Emissions (kgCO$_2$eq) |
|---|---|---|
| AlexNet on CIFAR-10 | 3.60 | 4.07 |
| ResNet-20 on CIFAR-10 | 10.24 | 11.58 |
| Pre ResNet-20 on CIFAR-10 | 9.26 | 10.48 |
| AllCNN-C on CIFAR-10 | 17.06 | 19.30 |
| PyramidNet 110a48 on CIFAR-10 | 73.86 | 83.53 |
| Wide ResNet 28-5 on CIFAR-10 | 42.90 | 48.52 |
| Wide ResNet 16-8 on CIFAR-10 | 36.08 | 40.80 |
| Wide ResNet 28-10 on CIFAR-10 | 105.30 | 119.10 |

Overall, experiment durations are short enough that TaylorGLO can be practically applied to different tasks to find customized loss functions.

