# OpenReview forum: "Optimizing Loss Functions Through Multivariate Taylor Polynomial Parameterization"
_ICLR.cc/2021/Conference — Reject_

### Official Review · AnonReviewer3 · 2020-10-26
**Useful idea, but evaluation could be improved.**

**Rating:** 5
**Confidence:** 4

**Review:**

This paper presents the TaylorGLO method for learning loss functions for classification.

## Pros
1. The paper is well written and quite easy to follow.
2. It proposes the novel idea of parameterising learned loss functions as Taylor polynomials, which overcomes the downside of previous work that relies on a slow two-stage optimisation process that first infers the structure of the loss function.
3. Networks trained with the learned loss functions typically outperform those trained with cross entropy.
4. Analysis is undertaken to determine why the learned loss functions work better than cross entropy (they penalise overly confident outputs), and also characterise when the loss functions discovered with TaylorGLO are most effective (better sample efficiency).

## Cons
1. The main performance comparison is with cross entropy, but there exist other methods for learning loss functions (e.g., GLO and [1]). The performance comparison with these methods should be more comprehensive: currently there is only a single comparison with GLO that uses a simple network trained on MNIST, which is not enough to make general conclusions about the relative performance of the two methods.
2. The potential for impact is limited. The main improvements in performance over cross entropy are seen when using older network architectures, but when using state of the art networks the improvements are much smaller. If it was shown that the learned loss functions could be transferred to new tasks, hence resulting in a modest performance increase without any additional computational overhead, then I would be inclined to revisit this point.
3. It is unclear what hypothesis is being tested when applying t-tests: given that the same data is used for each performance measurement it seems the hypothesis is that performance gains are robust to the choice of random seed. Hypothesis tests are typically used in machine learning to determine whether the experimental results will generalise to new tasks [2]. As they are currently presented, the results are likely to mislead some readers into inferring a false sense of generality of the results.

## Questions
1. Why use CMA-ES over gradient-based bi-level optimisation methods commonly used in meta-learning? Gradient-based optimisation is typically faster than evolutionary methods, and it seems like it would be possible to apply it in this situation.

[1] Sarah Bechtle et al. Meta-Learning via Learned Loss. arXiv:1906.05374, 2019.
[2] Janez Demšar. Statistical Comparisons of Classifiers over Multiple Data Sets. Journal of Machine Learning Research, 7(1):1−30, 2006.

---

> ### Author Response · Authors · 2020-11-25
> **Response**
>
> Dear Reviewer 3,
>
> Thank you for taking the time to review our paper. We would like to address your concerns as follows:
>
> **Cons:**
>
> *(On a single comparison with GLO:)* TaylorGLO was only compared to GLO on MNIST due to the exorbitant compute costs associated with running GLO for models that take longer. The scalability of TaylorGLO to deeper models is one key advantage over GLO.
>
> *(On impact with a broader range of settings:)* To demonstrate the efficacy of TaylorGLO more generally,  new experiments were included with different regularization techniques, an additional dataset, and an additional architecture:
> Settings:
> - CIFAR-10 AlexNet + Cutout
> - CIFAR-10 AlexNet + CutMix
> - CIFAR-10 AllCNN-C + CutMix
> - CIFAR-100 PyramidNet 110a48
> - CIFAR-100 PyramidNet 110a48 + Cutout
>
> Additionally, experiment durations and environmental impact have been added to Appendix G. Most experiments take tens of hours to run, making TaylorGLO practical in many settings, especially since it needs to be run only once for each architecture / dataset pair.
>
> The results are included in the main paper in Table 1. They all demonstrate significant improvements with TaylorGLO, strengthening the conclusions of the paper. Thanks for the suggestion!
>
> *(On hypotheses tested with t-test:)* The t-tests are included to show that the results are robust to all stochasticity in training. Generalization to different tasks is a different question, and not the focus of this current paper. The tested hypothesis is now clarified in Appendix A.5.
>
> **Question 1:**
>
> CMA-ES is a more general approach in that it allows non-differentiable objectives to be used to guide the search. In this paper, validation accuracy is itself used as an objective, but in future work we plan to extend the approach, additional objectives such as calibration.
>
> We hope these new analyses and the updates to the paper have addressed your concerns.
>
> Best regards,
> – The Authors

---

### Official Review · AnonReviewer4 · 2020-10-27
**Interesting idea, but not convincing enough**

**Rating:** 5
**Confidence:** 4

**Review:**

This paper proposed a method, called TaylorGLO, to learn the loss functions, for training deep neural network, by meta-learning. Specifically, the authors proposed to parameterize the loss function with multivariate Taylor polynomial, and then learn the parameters in the polynomial using evolutionary algorithm within the meta-learning framework. The experiments showed improved performance of the TaylorGLO over cross-entropy baseline on several datasets and with different network architectures.

Although the performance seems promising, there are several issues should be addressed:

1. My major concern is about the parameterization of the loss function. As is well known, a qualified loss function in machine learning generally contains only one valley concerning its geometric shape. However, with Taylor polynomial parameterization, a function might have multiple valleys, or even have no minimum.

2. The authors set the order of Taylor polynomial to 3, and simply claimed this is a better trade-off compared with other orders. However, this claim should be supported by empirical evaluations, i.e., use other orders to see the differences regarding the performance and evolution time.

3. Since the expensive evolutionary process is required to learn the loss function before really train the model, the computational cost should be shown and discussed in detail, especially that the performance improvement is not that much in some cases as shown in Table 1.

4. The comparison with GLO was only performed on MNIST dataset, where the accuracies between the two methods are very close.

---

> ### Author Response · Authors · 2020-11-25
> **Response**
>
> Dear Reviewer 4,
>
> Thank you for taking the time to review our paper. We would like to address your concerns as follows:
>
> (On function shape:) The experiments in this paper use third-order polynomials, and therefore the loss functions can indeed have at most one valley. More general TaylorGLO parameterizations, such as higher order polynomials, can in principle represent functions that are not useful loss functions. However, because the search space is smooth, CMA-ES is likely to quickly narrow in on areas with good parameterizations even in such settings.
>
> (On higher order approximations:) The decision to use third-order approximation was based on preliminary experiments where different orders from 2 to 6 were evaluated empirically, and third order found to strike the best balance between expressiveness and computational cost. Those results can be rerun and included in the final revised version if they are deemed helpful. For the current revision we did run a new experiment that evaluated the approximation ability of higher-order polynomials wrt. cross-entropy loss. The results, presented in a new Appendix E, demonstrate that higher order approximations are indeed better, and suggest that it might be useful to adjust the order of approximation to the task and architecture, as well as to the available computational resources.
>
> (On computational cost:) Experiment durations and environmental impact estimates have been added as a new Appendix G. Most experiments take only tens of hours to run, making TaylorGLO practical for many settings, especially since it needs to only be run once for each architecture / dataset pair.
> TaylorGLO was only compared to GLO on MNIST due to the exorbitant compute costs associated with running GLO for models that take longer. Indeed, the scalability of TaylorGLO to deeper models is one key advantage over GLO.
> We hope these new analyses and updates to the paper have addressed your concerns.
>
> Best regards,
> – The Authors

---

### Official Review · AnonReviewer2 · 2020-10-28
**Interesting idea and exciting area of work! How can we isolate effects of loss fn optimization vs. learning rate, etc?**

**Rating:** 6
**Confidence:** 2

**Review:**

Warning: I'm not an expert in loss function metalearning, so I've attempted to provide my perspective as someone interested in the topic but unfamiliar with prior work.

# Summary

This paper tackles the problem of loss function metalearning by proposing a novel parameterization of loss functions based on multivariate polynomial expressions.

The intuitive motivation is that any continuous and full differentiable loss function can be written as a $k$-th order multivariate Taylor expansion (i.e. a multivariate polynomial expression). So for high enough $k$, the space of such polynomials will be a reasonable approximation of the space of all loss functions we might care about.

The claimed advantages of the method are smoothness, lack of poles, compositionality, meaningful metric (closeness in parameter space ==> similar loss function), and tunable complexity (via $k$) of the search space.

The paper then proposes a loss function optimizer named TaylorGLO, which proposes to optimize loss functions in parameter space (i.e. coefficient space) using CMA-ES, which requires a continuous optimization space.

# Strengths

This is a neat idea, although I think calling it TaylorGLO is a bit of a misnomer as we're not starting with arbitrary functions that we're then approximating via Taylor expansion. Rather, the paper optimizes directly in polynomial space. Maybe "PolyGLO" is a more accurate name? The point can be made within the paper text that for sufficiently high $k$, the space of polynomials becomes an arbitrarily good approximation of any cont/diffble loss function thanks to Taylor's thm.

Regardless, I really like this approach! It definitely seems like a more promising approach than the referenced prior approach, which first optimizes the structure of the loss fn in discrete (tree) space and then optimizes the coefficients via continuous methods. The fact that the experiments found a function that outperforms "BaikalCMA" is welcome confirmation that we don't lose much by staying in continuous space.

# Weaknesses / unanswered questions

I'm worried that to some extent, the improvements from the new loss functions are thanks to an implicit learning rate tuning effect. The derivative of the loss function w.r.t. the model output directly scales the parameter updates for backpropogation-based SGD. So, maybe what's happening is that the learning rate is being tuned implicitly by TaylorGLO choosing a steeper or more compressed loss function. Maybe there's even a learning rate *scheduling* effect that's coming from TaylorGLO choosing loss functions with higher curvatures.

I think it's quite important to disentangle the effects of the loss function optimization from an implicit learning rate tuning, so I would ideally like to see experiments comparing TaylorGLO's best-accuracy loss function with plain old CE sweeped across learning rates and LR scheduling approaches. The difference in accuracy between TaylorGLO and a comprehensive CE sweep will be a more accurate indication of the effects of the loss optimization itself.

I'm also curious to know the extent of prior art in continuous loss function optimization. This approach seems like the first thing one would do if trying to design a loss function metalearning approach in continuous space, so I'd like to hear about what (if anything) has been done before.

# Overall Rating

I rate this paper a 6. I would be happy to see it in ICLR as is, but I think for its results to be exciting to the ICLR audience at large, the paper needs to convincingly isolate the effects of the function optimization itself.

# Comments

- p.1 "arbitrarily long, fixed-length vectors in a Hilbert space" seems self-contradicting. Is this meant to say "arbitrary norm"?
- Formatting of some of the \citep's can be improved, e.g. CMA-ES.
- I'd love to see a comparison of classification accuracy between standard cross-entropy and $k$-th order approximations of CE to get a sense of how much the fidelity loss actually matters.

---

> ### Author Response · Authors · 2020-11-25
> **Response**
>
> Dear Reviewer 2,
>
> Thank you for taking the time to review our paper. We would like to address your concerns as follows:
>
> **RE: Implicit learning rate tuning:**
> 	We appreciate this recommendation and have taken action. Indeed, we want to ensure that the improvement in performance is not solely due to an implicitly modified learning rate. In a new Appendix E, we provide a set of experiments that address this issue specifically. A sweep of learning rates with the cross-entropy loss shows that even with highly unstable learning rates that can improve accuracy in a handful of training sessions, the improvement is much smaller than that gained by training with a TaylorGLO loss function. We did not have time within the rebuttal period to analyze the effect of learning rate schedules, but we will continue the experiments and will include such analyses in the final submission.
>
> **RE: Accuracy of Taylor approximations of the cross-entropy loss function:**
> 	Thank you for this suggestion. We have now performed these analyses in Appendix F. As expected, higher-order approximations yield better performance, although all have accuracies that are a few percentage points lower than the actual cross-entropy loss. Thus, since TaylorGLO loss functions outperform the non-approximated cross-entropy loss, TaylorGLO discovers significantly different functions that are not merely approximations of the cross-entropy loss. Increasing the approximation order is an interesting (although computationally expensive) direction of future work, as is now mentioned in Appendix F.
>
> **RE: Small adjustments:**
> 	The recommendations on arbitrary norm and citep have been implemented in the revised paper. The related work section has been updated to point out that to our knowledge, TaylorGLO is the first method to utilize fixed-length vectors and continuous loss function optimization.
>
> We hope these new analyses and the updates to the paper have addressed your concerns.
>
> Best regards,
> – The Authors

---

> > ### Comment · AnonReviewer2 · 2020-11-25
> > **Thank you**
> >
> > Thank you for the revisions. Appendix F contains some really striking results! Order 3 results in 5% accuracy drop, and even order 5 has 3% accuracy loss. That is really surprising to me but useful to know.
> >
> > I maintain my rating as I would have liked to see a more rigorous analysis of the learning rate effects (the authors depict a learning rate sweep on a single task + suboptimal model architecture, and the learning rate scheduling effects are not addressed). Regardless, I think this is a paper that I would enjoy seeing in ICLR.

---

### Official Review · AnonReviewer1 · 2020-10-29
**This paper investigates the optimization of loss functions through multivariate Taylor expansion.**

**Rating:** 6
**Confidence:** 4

**Review:**

In this paper, the authors studied the optimization problem of loss functions, where the loss function is not referred to as the error criterion but the empirical error term in empirical risk minimization. The main approach proposed in this paper is the polynomial parametrization of the loss function via multivariate Taylor expansion, namely, equation (5), which should also be regarded as the main contribution of this paper.

Pros: The paper is well written and is easy to follow. The main approach and its rationality are well articulated. It is good to see that different approximates of loss functions are compared. The effectiveness of the proposed approach is also well illustrated numerically.

Cons: While I believe that the proposed approach works, it is not clear to me what is the relation between the newly proposed approach and the original approach where loss functions are used directly instead of their approximates. In my opinion, the new approach works not because of the approximate of the loss function but because of the new error criterion which is essentially a polynomial. This gives my main concern. I'm expecting more comments in this regard.

---

> ### Author Response · Authors · 2020-11-25
> **Response**
>
> Dear Reviewer 1,
>
> Thank you for taking the time to review our paper. We would like to clarify your concern:
>
> The goal of the TaylorGLO approach is not to use the Taylor expansion to approximate the loss function, instead, the Taylor polynomials  constitute a representation of the loss function that makes it possible to search for good loss functions effectively. In other words, the Taylor polynomials are not approximations of loss functions---they are loss functions themselves. How expressive these functions are is still an interesting question. To shed light on it, a new study is presented in Appendix F of the revised paper, analyzing the performance of Taylor approximations of the cross-entropy loss function. As expected, higher-order approximations yield better performance, although all have accuracies that are a few percentage points lower than the actual cross-entropy loss. This result suggests that higher order polynomials might be used to improve performance of TaylorGLO further, albeit with an increased computational cost. However, since third-order TaylorGLO loss functions already outperform the non-approximated cross-entropy loss, this result demonstrates that TaylorGLO discovers significantly different functions that are not merely approximations of the cross-entropy loss.
>
> We hope this new analysis and the updates to the paper have addressed your concern.
>
> Best regards,
> – The Authors

---

### Decision · Program_Chairs · 2021-01-07
**Final Decision**

**Decision:**

Reject

**Comment:**

Pros: Reviewers generally agreed the paper was well written and is easy to follow. The goal of learning loss functions also seems quite promising.

Cons: There were concerns about whether credit for experimental performance was attributable to the core algorithm+functional form presented in the paper. There was also some skepticism about the specific form of the learned loss. Of greatest concern, no reviewer argued for acceptance during discussion, and one reviewer lowered their score during discussion.